# Uncovering Capabilities of Model Pruning
# in Graph Contrastive Learning

## ABSTRACT

Graph contrastive learning has achieved great success in pre-training graph neural networks without ground-truth labels. Leading graph contrastive learning follows the classical scheme of contrastive learning, forcing model to identify the essential information from augmented views. However, general augmented views are produced via random corruption or learning, which inevitably leads to semantics alteration. Although domain knowledge guided augmentations alleviate this issue, the generated views are domain specific and undermine the generalization. In this work, motivated by the firm representation ability of sparse model from pruning, we reformulate the problem of graph contrastive learning via contrasting different model versions rather than augmented views. We first theoretically reveal the superiority of model pruning in contrast to data augmentations. In practice, we take original graph as input and dynamically generate a perturbed graph encoder to contrast with the original encoder by pruning its transformation weights. Furthermore, considering the integrity of node embedding in our method, we are capable of developing a local contrastive loss to tackle the hard negative samples that disturb the model training. We extensively validate our method on various benchmarks regarding graph classification via unsupervised and transfer learning. Compared to the state-of-the-art (SOTA) works, better performance can always be obtained by the proposed method.

## CCS CONCEPTS

• **Computing methodologies → Artificial intelligence**; • **Mathematics of computing → Graph algorithms**.

## KEYWORDS

Graph Contrastive Learning, Model Pruning, Graph Classification

## 1 INTRODUCTION

In light of the depletion of labeled data and the hardness of annotation, plenty of research attention has been moved from supervised learning to unsupervised learning [6, 14]. While in the graph domain, the same issue exists [17]. Correspondingly, referring to the design for unlabeled data training in natural language processing [48] and computer vision [14], several solutions have been dug out through plenty of research efforts and collectively called

graph contrastive learning, such as GraphCL [51], AD-GCL [36] and RGCL [24].

In general, graph contrastive learning sticks to the twin-tower architecture of contrastive learning [4, 43], in which two augmented views are generated from the input graph, and the model loss (i.e., NT-Xent loss [4]) maximizes the mutual information between the two output embeddings of the two augmented views. With this design, the trained model is capable of capturing the essential information of graphs [26]. Moreover, researchers have found that the quality of views influences the performance of contrastive learning models [38]. Therefore, plenty of research efforts have been devoted to the generation of effective views that lead to better performance for graphs through the data augmentation [36, 50].

Contrastive data augmentation on graphs presents a significant challenge compared to images due to the complex structural information and diverse contexts inherent in graph data [8]. Inspecting prior studies on graph contrastive learning, we can systematize the two common paradigms for view generation. The first category is the random or learnable data corruption, such as the four types of general augmentations (node dropping, edge perturbation, attribute masking, and subgraph) in GraphCL [51] and learnable edge dropping in AD-GCL [36]. Despite the effectiveness of these graph views on various tasks, the proposed data augmentations via random corruption or learning suffer from structural damage and artificially introduced noise, which could alter the fundamental property of input graphs. Based on predefined sub-structure substitution rules [35] or contrasted with 3-dimensional geometric views [25, 32], the second way is to integrate the domain knowledge to alleviate the issue of semantic alteration with data corruption. However, the fusion of domain knowledge inevitably undermines the generalization of the pre-trained model to other domains [24].

To fully leverage the potential of contrastive learning in the graph domain, it is desirable to develop a graph contrastive learning model that can preserve semantic information while remaining independent of domain-specific knowledge. To address this challenge, we shift our focus from contrastive views to the graph encoder within the contrastive learning framework. Drawing inspiration from model compression techniques, we note that the performance of sparse sub-networks could be comparable to their complete versions [5, 12], suggesting that pruning may be a viable approach for graph contrastive learning. Accordingly, we introduce a novel framework called LAMP that enables Graph Contrastive **L**earning vi**A M**odel **P**runing to remedy these issues in previous works.

The framework of LAMP is shown in Figure 1. LAMP takes the original graph as model input to prevent semantic information alteration from graph corruption. While fostering the model with the ability to identify the essential information, we employ pruning [12, 15] for perturbation. In particular, since the pruned graph

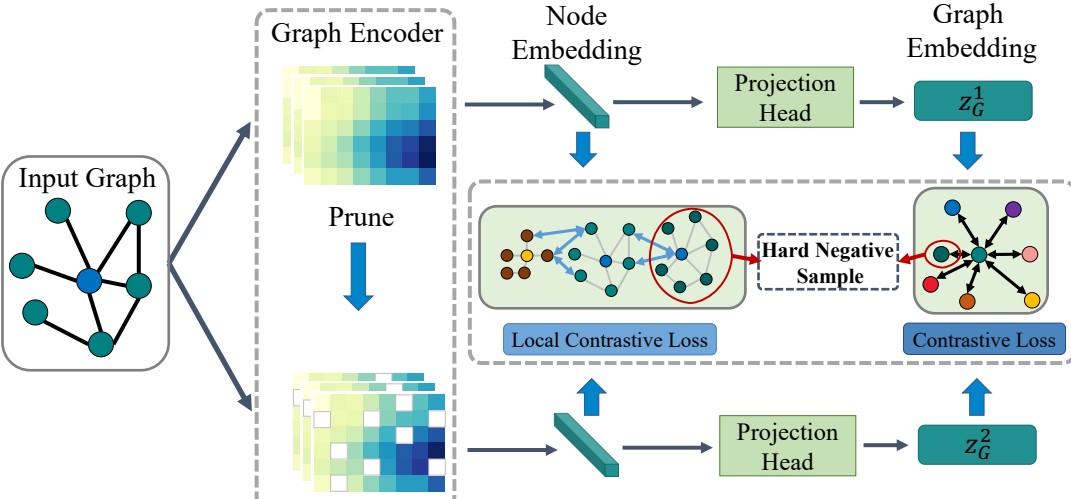

**Figure 1: Framework of LAMP. One branch takes the original graph as input instead of the augmented view. The other branch is pruned from the former online and also embeds the original graph. Besides the ubiquitous NT-Xent loss, the graph encoder is jointly optimized with a local contrastive loss to optimize the hypersphere of contrastive learning.**

encoder is always obtained from the latest encoder, the two contrastive embeddings will co-evolve, which ensures model convergence. Moreover, considering the hard negative samples that have different structural properties but output similar graph embedding, we develop a novel loss to enhance the contrastive learning with the node embeddings. Despite the simplicity, coupling the two strategies together enable us to perform effective contrastive learning on graphs with model perturbation. Besides the superior performance of LAMP in the extensive experiments of unsupervised and transfer learning for graph classification, we also theoretically explain the superiority of model pruning compared to data augmentation. The emerging contributions are elaborated below:

- We reformulate the framework of graph contrastive learning from model compression, which allows the model pretraining free from semantic information alteration and profound domain knowledge fusion.
- Convinced by the capability of pruning in representation learning, we theoretically analyze the superiority of model pruning compared to data augmentation and present the instantiation, called LAMP.
- Considering the correspondence of node embeddings in positive pair samples, we further enhance LAMP with the ability of handling hard negative samples by a local contrastive loss.
- LAMP suppresses the SOTA competitors through extensive experiments in unsupervised and transfer learning.

## 2 RELATED WORKS

Graph contrastive learning has been broadly adopted for tasks without ground-truth labels, such as graph classification [24, 51], node classification [8], and link prediction [54]. Here, we devote our attention to contrastive learning for graph classification, the most relevant topic of this work.

### 2.1 Graph Contrastive Learning

Great success has been achieved by graph contrastive learning in self-supervised graph representation learning. Among various research, the study of the contrastive view is a key issue in graph contrastive learning. Recently, based on the data augmentation on images, numerous works have explored the feasibility of augmentations on graphs [24, 36, 50–52]. While gratifying, the proposed data augmentations via random perturbation or learning suffer from structural damage and noisy information [36, 51]. To tackle this issue, several works have attempted to preserve the graph semantic structure by resolving profound domain knowledge into augmentations, such as MoCL [35], KCL [7], GraphMVP [25] and 3D-Infomax [32] However, this domain knowledge only exists on molecules, which significantly limits the generality. Moreover, beyond the general contrastive learning framework, DGCL [23] disentangles the graph encoder, and OEPG [47] explores the semantic structure of datasets. Although they present excellent performance, they still rely on the graph augmentations and thus are orthogonal to these works that keep the general contrastive learning framework; put differently, other models can work with the framework of DGCL and OEPG to produce superior performance. Among recent works, SimGRACE [44] preserves semantics by disturbing the model weight with Gaussian noise. However, the introduced noise is data-agnostic, which could degenerate the model performance when the actual data distribution goes beyond the Gaussian distribution and explain the sub-optimal performance of SimGRACE in the experiment section.

### 2.2 Model Pruning

In the early stage, pruning is generally a technique to improve model efficiency, which aims to shrink model size at surprisingly little sacrifice of model performance [20], and various pruning techniques have been proposed and effective for that goal [12, 15, 22]. Recently,

besides the inherent function of model compression, several works explored its deeper connection with model generalization [9, 53]. Moreover, the capability of pruning on model memorization has also been validated with long-tail distribution dataset [18]. In this paper, we particularly investigate contrastive learning via model pruning for graph representation learning. Note that, model pruning is first employed in this work to address the issue of semantics alteration caused by data augmentation.

## 3 NOTATIONS AND PRELIMINARIES

Before the elaboration of LAMP, here, some preliminary concepts and notations are given. Let $\mathcal{E}$ and $\mathcal{V}$ be the sets of edges and vertices, a simple graph can be formally written as $G = (\mathcal{V}, \mathcal{E})$.

**Graph representation learning.** Generally, GNNs based on a message-passing scheme serve as the graph encoder [10]. A GNN learns an embedding $h_v \in \mathbb{R}$ for each node and a vector $h_G \in \mathbb{R}$ is produced by a READOUT function for graph $G$. For an $L$-layer GNN, each node vector is decorated with the $L$-hop information from its neighbors. The hidden vector of node $v$ in layer $c$ can be obtained by:

$$h_v^{(c)} = f_U^{(c)}(h_v^{(c-1)}, f_M^{(c)}(\{(h_v^{(c-1)}, h_u^{(c-1)})|u \in N(v)\})), \quad (1)$$

where $f_U^{(c)}$ aims to update each node vector in current layer, $f_M^{(c)}$ is the designed function for message-passing on graphs, the first-order neighbor nodes of $v$ is represented as $N(v)$, and $h_v^{(c)}$ denotes the hidden feature of $v$ in the $c$-th layer. After $L$ iterations, the entire graph representation can be written as

$$h_G = f_R(\{h_v|v \in \mathcal{V}\}), \quad (2)$$

where $f_R$ pools the final set of node representations and is generally a summation or averaging function.

**Graph contrastive learning.** A typical unsupervised model via contrastive learning takes two views from one graph as input, and the two views are produced by two data augmentation operators and serve as a positive pair. At the phase for pre-training, a GNN-based encoder is used for structural information modeling of the input views, and a projection head is further employed to embed the two views into the same feature space for contrast. Output feature vectors $h_i^1$ and $h_i^2$ from the same graph are expected to identify themselves from the others. Thus, the NT-Xent loss [4] is adopted to achieve this goal via maximizing the consensus of a positive pair:

$$\mathcal{L}_i = -\log \frac{e^{sim(h_i^1, h_i^2)/\tau}}{\sum_{j=1, j \neq i}^N e^{sim(h_i^1, h_j^2)/\tau}}, \quad (3)$$

where $N$ is the batch size, $\tau$ controls the temperature parameter, and $sim(h^1, h^2)$ generally refers to a cosine similarity function $\frac{h^{1\top}h^2}{||h^1|| \cdot ||h^2||}$. In particular, there are two types of negative pairs; put differently, $h_i^1$ can pair with all $h_j^2$, and $h_i^2$ can pair with all $h_j^1$.

**The mutual information maximization principle.** Graph contrastive learning leverages the principle of mutual information maximization (InfoMax) to enhance the correspondence between a graph representation and its corresponding views from various augmentations. The graph representation $h_G$ is supposed to contain the feature underlying $G$, because the representation is expected

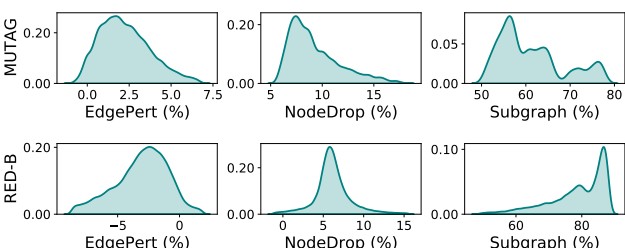

**Figure 2: Quantification of structural damage from data augmentation. Percent change in structural entropy of MUTAG and REDDIT-BINARY after data augmentation (i.e., Edge perturbation, Node dropping, and Subgraph with 20% strength from GraphCL).**

to distinguish the graph $G$ from others within the same batch. The principle of mutual information maximization can formally be

$$\text{InfoMax:} \quad \max I(G; h_G), where \ G \sim \mathbb{P}_G, \quad (4)$$

where $\mathbb{P}_G$ denotes the distribution defined over the graph $G$ and $I(\cdot)$ refers to the mutual information.

## 4 METHODOLOGY

In this section, by turning the attention from data augmentation to model pruning, we bring about the proposed graph contrastive learning framework, termed *LAMP*. Before the detailed description, we first give the motivation from the quantification of structural damage caused by general graph augmentation.

### 4.1 Quantification of Structural Damage from Data Augmentation

In previous works regarding graph contrastive learning, data augmentation is a general technique to help the graph encoder identify the essential information of input graphs [8, 49, 51]. Despite various forms, they are mostly built upon the concept of topology corruption, such as node dropping, edge perturbation, and learnable graph generation. Although these works have shown some extent of effectiveness on the actual tasks, the underlying structural damage remains, which inevitably leads to semantics alteration and undermines the model performance.

Now, based on structural entropy [21], a metric for structural information, we are the first to give the quantitative illustration of structural damage caused by data augmentations. Give a simple graph $G = (\mathcal{V}, \mathcal{E})$, the structural information underlying $G$ can be measured by:

$$\mathcal{H}(G) = -\sum_{v \in \mathcal{V}} \frac{g_v}{vol(\mathcal{V})} \log \frac{g_v}{vol(\mathcal{V})}, \quad (5)$$

in which $vol(\mathcal{V})$ denotes the sum of all node degrees and $g_v$ is the node degree of $v$. Among various forms of data augmentation proposed in previous works, here, we give the quantitative illustration of structural damage by three ubiquitous rules from GraphCL [51], including subgraph, edge perturbation, and node dropping. In particular, the augmentation strength is consistent with the setting in

GraphCL, that is, 20%. The structural damage of graph $G$ is measured by the percent change of its structural entropy. Formally, given any data augmentation function $t$, the percent change of structural entropy will be

$$\mathcal{L}_{SE} = 1 - \frac{\mathcal{H}(t(G))}{\mathcal{H}(G)}. \tag{6}$$

The quantitative illustrations of structural damage caused by three data augmentation rules on a social network dataset (i.e., REDDIT-BINARY) and a bioinformatic dataset (i.e., MUTAG) are shown in Figure 2. [1] The effect of structural damage varies with the augmentation rules. Specifically, node dropping and subgraph lead to different degrees of structural damage, and the information loss caused by subgraph is the largest and generally over 50%. Besides the simple information loss, the structure damage composition of edge perturbation is more complex; put differently, in light of the distribution of REDDIT-BINARY, edge perturbation even introduces external data noise with the additional edges, which further interferes with the model from learning the actual structural information. Therefore, we naturally wonder: Can we design a more advanced model with effective contrastive pairs without structural damage? Next, to tackle our expectations, we present the superiority of model pruning in contrast to data augmentation.

## 4.2 Theoretical Superiority of Model Pruning

Through theoretical analysis, in this subsection, we present the property of graph encoder trained from model pruning.

THEOREM 4.1. *Suppose the graph encoder $f$ is implemented by a GNN with at least 2 layers and $f^*$ is the optimal version. Given a general data augmentation function $t$, the optimal pruned encoder $f_p^*$ satisfies,*

1. *$I(f_p^*(G); Y) \geq I(f^*(t(G)); Y)$;*
2. *$I(f_p^*(G); f^*(G)) \geq I(f^*(t^1(G)); f^*(t^2(G)))$.*

Statement 1 in Theorem 4.1 guarantees a lower bound of the mutual information between the learned representations and the labels of the downstream task; put differently, the learned essential information via the sparse encoder is more than views from augmentations.

Statement 2 in Theorem 4.1 suggests that the training performance of LAMP is better than the models based on augmentations in the architecture of graph contrastive learning.

PROOF. Suppose $\mathbf{G}$ is a set of graphs. According to the definition, $f^* = argmax_f I(f(G); G)$, $f^*$ should be injective. Given some graph $G \in \mathbf{G}$, $G \Rightarrow f^*(G)$ is an injective deterministic mapping. Thus, for any random variable $Q$,

$$I(f^*(G); Q) = I(G; Q). \tag{7}$$

When there is $Q = Y$, we will have,

$$I(f^*(G); Y) = I(G; Y). \tag{8}$$

Moreover, in light of theoretically proof in [27], a depth-two network can be approximated by pruning a random-weighted subnetwork $f_p^*$ as follows:

$$\sup_{G \in \mathbf{G}} |f^*(G) - f_p^*(G)| \leq \epsilon. \tag{9}$$

Accordingly, we have the following proof:

$$
\begin{aligned}
& I(f_p^*(G); Y) - I(f^*(G); Y) \\
=& H(f_p^*(G)) + H(Y) - H(f_p^*(G), Y) \\
& - H(f^*(G)) - H(Y) + H(f^*(G), Y) \\
=& H(f_p^*(G)) - H(f^*(G)) \\
& - (H(f_p^*(G), Y) - H(f^*(G), Y)) \\
\overset{(a)}{\leq}& 2\epsilon^2, 
\end{aligned} \tag{10}
$$

where $(a)$ is because of the arbitrariness of $\epsilon$ and the continuity of the entropy $H(\cdot)$. Meanwhile, because $\epsilon$ can be arbitrarily small, we can achieve

$$I(f_p^*(G); Y) = I(f^*(G); Y). \tag{11}$$

Now, introducing the data processing inequality [37] for data augmentation,

$$
\begin{aligned}
I(f^*(G); Y) &= I(G; Y) \\
&\geq I(t(G); Y) \\
&= I(f^*(t(G)); Y). 
\end{aligned} \tag{12}
$$

Combining above equations, we have the statement 1:

$$I(f_p^*(G); Y) = I(f^*(G); Y) \geq I(f^*(t(G)); Y). \tag{13}$$

Next, we come to proof the second statement,

$$
\begin{aligned}
& I(f^*(t^1(G)); f^*(t^2(G))) \\
=& I(f^*(t^1(G)); (f^*(t^2(G)), Y)) \\
& - I(f^*(t^1(G)); (Y|f^*(t^2(G)))) \\
\leq& I(f^*(t^1(G)); (f^*(t^2(G)), Y)) \\
=& I(f^*(t^1(G)); Y) \\
& + I(f^*(t^1(G)); (f^*(t^2(G))|Y)). 
\end{aligned} \tag{14}
$$

Then, according to the data processing inequality [37], we move forward to

$$
\begin{aligned}
& I(f^*(t^1(G)); Y) + I(f^*(t^1(G)); (f^*(t^2(G))|Y)) \\
\leq& I(f^*(G); Y) + I(f_p^*(t^1(G)); (f^*(t^2(G))|Y)) \\
\leq& I(f^*(G); Y) + I(f_p^*(G); (f^*(G)|Y)) \\
=& I(f_p^*(G); Y) + I(f_p^*(G); (f^*(G)|Y)) \\
=& I(f_p^*(G); (f^*(G), Y)). 
\end{aligned} \tag{15}
$$

Finally, for the reason of $f_p^*(G) \perp_{f^*(G)} Y$,

$$
\begin{aligned}
& I(f_p^*(G); (f^*(G), Y)) \\
=& I(f_p^*(G); (f^*(G), Y)) - I(f_p^*(G); (Y|f^*(G))) \\
=& I(f_p^*(G); f^*(G)), 
\end{aligned} \tag{16}
$$

which concludes the proof of the statement 2. □

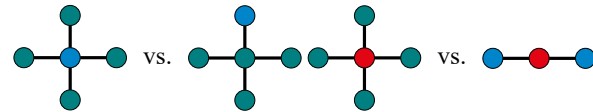

(a) Same structure but different (b) Different structure and different node features.

Figure 3: Illustration of hard negative samples. Via contrasting the graph embeddings, the pre-trained model is hard to distinguish this two kinds of graphs.

## 4.3 Instantiation of LAMP

With the superiority of pruning in graph contrastive learning, now, we move from theory to practice. Instead of producing contrastive pairs via data augmentation, here, we give our model design via model pruning. Figure 1 general pictures the workflow of LAMP. As can be seen, LAMP sticks to the twin-tower architecture of GraphCL [51], while gets rid of its trail-and-error data augmentation rules. Especially, LAMP feeds the original graph into the dense graph encoder and its pruned version to contrast their embeddings. Formally, let $G = (\mathbf{A}, \mathbf{X})$ be the input graph, $\mathbf{A}$ is the adjacency matrix, and $\mathbf{X}$ is the initial node features. For the $l$-th layer of graph encoder, the node representations of $G$ via Equation 1 will be:

$$H^{(l)} = f_U^{(l)}(f_M^{(l)}(\mathbf{A}, H^{(l-1)}); W^{(l)}), \qquad (17)$$

$$H_p^{(l)} = f_U^{(l)}(f_M^{(l)}(\mathbf{A}, H_p^{(l-1)}); p(W^{(l)})), \qquad (18)$$

where $W^{(l)}$ is the weight matrix of update function $f_U^{(l)}$ and $p(\cdot)$ is the pruning function on model weight $W$.

In practice, we prune the dense graph encoder according to a predefined ratio $\gamma$ with the given pruning strategy, that is, the weight values of $W^l$ will be masked if they are ranked below $\gamma$. In particular, the sparse degree of the graph encoder can be controlled by tuning the pruning ratio $\gamma$, which meets various demands of different datasets. Detailed discussion about the pruning ratio is conducted in the ablation study. Moreover, to avoid drastic gradient changes and save computations, the wight matrix will be pruned at the beginning of each epoch. Since the sparse encoder is always obtained and updated from the latest dense version, the two branches will co-evolve during training.

**Remark.** To avoid possible confusion, we emphasize that the adoption of pruning is not aimed at enhancing model efficiency, but rather at boosting the performance of contrastive learning. Furthermore, our approach in this study is not tied to a specific pruning strategy, but is compatible with any given pruning methods. Specifically, we have implemented our method using two distinct pruning techniques: magnitude pruning [12] and soft filter pruning [15], which are referred to as **LAMP-Mag** and **LAMP-Soft**, respectively.
**Projection head.** After obtaining the graph representations via global pooling, a projection head $g(\cdot)$ is employed to cast the representations to another feature space for contrasting. In the framework of LAMP, a two-layer perceptron is also employed to produce the graph final representations $z_G^1$ and $z_G^2$,

$$z_G^1 = g(h_G^1); \quad z_G^2 = g(h_G^2). \qquad (19)$$

---

**Algorithm 1** Pre-training algorithm of LAMP

**Input:** the training dataset $\mathbb{G} = \{G_1, G_2, \cdots\}$, graph encoder $f(\cdot)$ with weight matrix $\mathbf{W}_f$, projection head $g(\cdot)$ with weight matrix $\mathbf{W}_h$, pruning ratio $\gamma$ and learning rate $r$.

1: Initialize graph encoder $f(\cdot)$.
2: **for** each epoch **do**
3:    Perform pruning to get $f_p(\cdot)$ with ratio $\gamma$;
4:    **for** each mini-batch **do**
5:       Obtain the node representations $\mathbf{H}_\mathcal{V}^1$ and $\mathbf{H}_\mathcal{V}^2$ through the two graph encoders $f(\cdot)$ and $f_p(\cdot)$;
6:       Obtain the graph representations $\mathbf{Z}^1$ and $\mathbf{Z}^2$ through Equation 2 and projection head $g(\cdot)$;
7:       Calculate batch loss $\mathcal{L}$ based on Equation 21;
8:       Update weights:
$$\mathbf{W}_f' \leftarrow \mathbf{W}_f - r\nabla_{\mathbf{W}_f}\mathcal{L}$$
$$\mathbf{W}_g' \leftarrow \mathbf{W}_g - r\nabla_{\mathbf{W}_g}\mathcal{L}$$
9:    **end for**
10: **end for**

---

## 4.4 Hard Negative Samples

Among the research on contrastive learning, hard negative samples are quite ubiquitous and have great potential to improve model performance[29]. However, little attention has been drawn to the hard negative samples within current contrastive learning for graph classification. As shown in Figure 3, there are two kinds of hard negative samples. In Figure 3a, this negative pair has the same topology but different features. Let $h_{color}$ ($g$ for green, $b$ for blue) denote the node features, the graph representations would be similar after pooling as $f_R(4 \times h_g + f_b) = f_R(4 \times h_g + f_b)$. In Figure 3b, this negative pair has different topology and node features. Let $h_b = 2h_g$, the graph representations would be also similar through summation or maximization pooling function. Given the common design of current contrastive learning for graph classification, the model is encouraged to enlarge the distance of negative pairs via graph representations, which may fail with the two kinds of negative samples in Figure 3 and deteriorate model performance.

Despite the hardness of distinguishing the two kinds of negative samples from the graph representations, we may be able to spot some opportunities from the local features. For example, the green nodes could be effortlessly identified from the blue nodes in Figure 3a, and the read nodes also have obvious differences compared to the green and blue nodes in Figure 3b. Therefore, in order to cultivate the ability of contrastive learning to tackle hard negative samples, we propose a local contrastive loss with the node embeddings.
**Local Contrastive Loss.** Because each node embedding matrix after graph encoder contains the full set of nodes in original graph, here, we propose a local contrastive loss to enhance graph learning from the node level. Our critical insight is that current models lack the ability to separate the hard negative samples via graph representations, while a contrastive angle based on node representations would be helpful in this scene. Specifically, as the general NT-Xent loss enforces the dissimilarity among different graphs, we move forward to generalizing this dissimilarity to nodes. Formally, the

Table 1: Average accuracies (%) ± Std. of LAMP and compared methods under the setting of unsupervised learning. Bold indicates the best performance over all methods. Underline represents the second best. A.A. refers to the average accuracy over eight benchmarks. A.R. implies the abbreviation of average rank. The results of baselines are derived from the published works and - indicates the data missing in the such works.

| | NCI1 | PROTEINS | DD | MUTAG | COLLAB | RED-B | RED-M5K | IMDB-B | A.A. | A.R. |
|---|---|---|---|---|---|---|---|---|---|---|
| GL | - | - | - | 81.66±2.11 | - | 77.34±0.18 | 41.01±0.17 | 65.87±0.98 | - | 15.5 |
| WL | 80.01±0.50 | 72.92±0.56 | - | 80.72±3.00 | - | 68.82±0.41 | 46.06±0.21 | 72.30±3.44 | - | 12.5 |
| DGK | 80.31±0.46 | 73.30±0.82 | - | 87.44±2.72 | - | 78.04±0.39 | 41.27±0.18 | 66.96±0.56 | - | 11.9 |
| node2vec | 54.89±1.61 | 57.49±3.57 | - | 72.63±10.20 | - | - | - | - | - | 16.7 |
| sub2vec | 52.84±1.47 | 53.03±5.55 | - | 61.05±15.80 | - | 71.48±0.41 | 36.69±0.42 | 55.26±1.54 | - | 17.5 |
| graph2vec | 73.22±1.81 | 73.30±2.05 | - | 83.15±9.25 | - | 75.78±1.03 | 47.86±0.26 | 71.10±0.54 | - | 13.6 |
| MVGRL | - | - | - | 75.40±7.80 | - | 82.00±1.10 | - | 63.60±4.20 | - | 15.7 |
| InfoGraph | 76.20±1.06 | 74.44±0.31 | 72.85±1.78 | 89.01±1.13 | 70.65±1.13 | 82.50±1.42 | 53.46±1.03 | 73.03±0.87 | 74.02 | 9.8 |
| GraphCL | 77.87±0.41 | 74.39±0.45 | 78.62±0.40 | 86.80±1.34 | 71.36±1.15 | 89.53±0.84 | 55.99±0.28 | 71.14±0.44 | 75.71 | 8.9 |
| JOAO | 78.07±0.47 | 74.55±0.41 | 77.32±0.54 | 87.35±1.02 | 69.50±0.36 | 85.29±1.35 | 55.74±0.63 | 70.21±3.08 | 74.75 | 10.6 |
| JOAOv2 | 78.36±0.53 | 74.07±1.10 | 77.40±1.15 | 87.67±0.79 | 69.33±0.34 | 86.42±1.45 | 56.03±0.27 | 70.83±0.25 | 75.01 | 9.6 |
| AD-GCL | 73.91±0.77 | 73.28±0.47 | 75.79±0.87 | 88.74±1.85 | 72.02±0.56 | 90.07±0.85 | 54.33±0.32 | 70.21±0.68 | 74.79 | 10.2 |
| AutoGCL | 82.00±0.29 | 75.80±0.36 | 77.57±0.60 | 88.64±1.08 | 70.12±0.68 | 88.58±1.49 | 56.75±0.18 | 73.30±0.40 | 76.59 | 6.0 |
| RGCL | 78.14±1.08 | 75.03±0.43 | 78.86±0.48 | 87.66±1.01 | 70.92±0.65 | 90.34±0.58 | 56.38±0.40 | 71.85±0.84 | 76.15 | 6.6 |
| SimGRACE | 79.12±0.44 | 75.35±0.09 | 77.44±1.11 | 89.01±1.31 | 71.72±0.82 | 89.51±0.89 | 55.91±0.34 | 71.30±0.77 | 76.17 | 6.9 |
| SEGA | 79.00±0.72 | 76.01±0.42 | 78.76±0.57 | 90.21±0.66 | 74.12±0.47 | 90.21±0.65 | 56.13±0.30 | 73.58±0.44 | 77.25 | 4.3 |
| GCS | 77.37±0.30 | 75.02±0.39 | 77.22±0.30 | 90.45±0.81 | 75.56±0.41 | **92.98±0.28** | 57.04±0.49 | 73.43±0.38 | 77.39 | 5.0 |
| LAMP-Mag | **82.62±0.31** | 76.75±0.67 | 79.47±0.97 | 90.02±1.59 | 74.62±0.75 | 90.58±0.46 | 56.42±0.26 | 73.46±0.65 | 77.99 | 2.8 |
| LAMP-Soft | 82.17±0.48 | **77.34±0.53** | **80.03±0.85** | **90.89±1.04** | **75.96±0.67** | 91.63±0.55 | **57.38±0.41** | **75.14±0.59** | **78.82** | **1.3** |

local contrastive loss can be formulated as:

$$\mathcal{L}_{LocalC}^{v_i} = -\log \frac{e^{sim(h_{v_i}^1, h_{v_i}^2)/\tau}}{\sum_{\hat{G} \neq G}^{\mathbb{G}} \sum_{j=1}^{|\mathcal{V}_{\hat{G}}|} e^{sim(h_{v_i}^1, \hat{h}_{v_j}^2)/\tau}}, \quad (20)$$

where $\mathbb{G}$ refers to a training batch. Note that the local contrastive loss also has two kinds of negative pairs. In particular, since the node batch size varies among different datasets, LAMP will randomly sample a sub-node set of size $N_s$ for large dataset to avoid high computation costs. Considering the batch node size in actual model training, $N_s$ is fixed to 5,000 in the experiment setting.

**Remark.** Although the local contrastive loss for hard negative samples is relatively simple in its presentation, it is not easy to implement when the two graphs of a positive pair have different structures due to corruption. This limitation may account for why previous graph contrastive learning models have overlooked this approach and consequently produced sub-optimal performance.

Currently, we have presented the main components of the proposed LAMP that aims to help graph contrastive learning free from structural damage caused by data augmentation and enhance model training facing hard negative samples. The final objective function of LAMP for pre-training is

$$\min \mathcal{L} = \mathcal{L}_G + \alpha \mathcal{L}_{LocalC}, \quad (21)$$

where $\mathcal{L}_G$ is the NT-Xent loss, $\alpha$ controls the loss weight. Algorithm 1 reveals the pre-training procedure of LAMP.

## 5 EXPERIMENTS

In this section, we are devoted to evaluating LAMP with extensive experiments [2]. Under the setting of unsupervised and transfer learning, LAMP empirically shows its superiority compared to the SOTA competitors. Ablation studies regarding hyper-parameters are further conducted to make an in-depth analysis.

### 5.1 Experimental Setup

**Datasets.** For unsupervised learning, various benchmarks are adopted from TUDataset [28], including COLLAB, REDDIT-BINARY, REDDIT-MULTI-5K, IMDB-BINARY, NCI1, MUTAG, PROTEINS and DD. For transfer learning, ZINC15 [33] dataset is adopted for pre-training. In particular, a subset with two million unlabeled molecular graphs are sampled from the ZINC15. We employ the eight ubiquitous benchmarks from the MoleculeNet dataset [42] as the downstream experiments regarding transfer learning. Further details are shown in Appendix B.

**Learning protocol.** Following the learning setting in GraphCL [51], the corresponding learning protocols are adopted for a fair comparison. (a) In unsupervised representation learning, all data is used for model pre-training and the learned graph embeddings are then fed into a non-linear SVM classifier to perform 10-fold cross-validation. (b) In transfer learning, we first pre-train the model on ZINC15. Then, we finetune and evaluate the model on MoleculeNet dataset using the scaffold split scheme [3].

**Configuration.** To keep in line with GraphCL [51], the same GNN architectures are employed with their original hyper-parameters under individual experiment settings. Specifically, in unsupervised

---
[2]The code of LAMP will be public after acceptance.

**Table 2: Average test ROC-AUC (%) ± Std. of LAMP along with baselines on eight downstream benchmarks under the setting of transfer learning. The results of baselines are derived from the corresponding works. Bold indicates the best performance among all baselines. Underline gives the second best. Avg. shows the average ROC-AUC over all datasets. A.R. implies the abbreviation of average rank and - indicates the data missing in the such works.**

| | BBBP | Tox21 | ToxCast | SIDER | ClinTox | MUV | HIV | BACE | Avg. | A.R. |
|---|---|---|---|---|---|---|---|---|---|---|
| No Pre-Train | 65.8±4.5 | 74.0±0.8 | 63.4±0.6 | 57.3±1.6 | 58.0±4.4 | 71.8±2.5 | 75.3±1.9 | 70.1±5.4 | 66.95 | 17.5 |
| Infomax | 68.8±0.8 | 75.3±0.6 | 62.7±0.4 | 58.4±0.8 | 69.9±3.0 | 75.3±2.5 | 76.0±0.7 | 75.9±1.6 | 70.29 | 15.3 |
| EdgePred | 67.3±2.4 | 76.0±0.6 | 64.1±0.6 | 60.4±0.7 | 64.1±3.7 | 74.1±2.1 | 76.3±1.0 | 79.9±0.9 | 70.28 | 12.7 |
| AttrMasking | 64.3±2.8 | 76.7±0.4 | 64.2±0.5 | 61.0±0.7 | 71.8±4.1 | 74.7±1.4 | 77.2±1.1 | 79.3±1.6 | 69.90 | 12.3 |
| ContextPred | 68.0±2.0 | 75.7±0.7 | 63.9±0.6 | 60.9±0.6 | 65.9±3.8 | 75.8±1.7 | 77.3±1.0 | 79.6±1.2 | 70.89 | 11.3 |
| GraphMAE | 72.0±0.6 | 75.5±0.6 | 64.1±0.3 | 60.3±1.1 | 82.3±1.2 | 76.3±2.4 | 77.2±1.0 | 83.1±0.9 | 73.85 | 8.5 |
| GraphMVP | 68.5±0.2 | 74.5±0.4 | 62.7±0.1 | 62.3±1.6 | 79.0±2.5 | 75.0±1.4 | 74.8±1.4 | 76.8±1.1 | 71.70 | 14.1 |
| GraphCL | 69.68±0.67 | 73.87±0.66 | 62.40±0.57 | 60.53±0.88 | 75.99±2.65 | 69.80±2.66 | 78.47±1.22 | 75.38±1.44 | 70.77 | 14.9 |
| JOAO | 70.22±0.98 | 74.98±0.29 | 62.94±0.48 | 59.97±0.79 | 81.32±2.49 | 71.66±1.43 | 76.73±1.23 | 77.34±0.48 | 71.89 | 13.9 |
| JOAOv2 | 71.39±0.92 | 74.27±0.62 | 63.16±0.45 | 60.49±0.74 | 80.97±1.64 | 73.67±1.00 | 77.51±1.17 | 75.49±1.27 | 72.12 | 12.9 |
| LP-Info | 71.40±0.55 | 74.54±0.45 | 63.04±0.30 | 59.70±0.43 | 74.81±2.73 | 72.99±2.28 | 76.96±1.10 | 80.21±1.36 | 71.71 | 13.4 |
| AD-GCL | 70.01±1.07 | 76.54±0.82 | 63.07±0.72 | 63.28±0.79 | 79.78±3.52 | 72.30±1.61 | 78.28±0.97 | 78.51±0.80 | 72.72 | 10.3 |
| AutoGCL | 73.36±0.77 | 75.69±0.29 | 63.47±0.38 | 62.51±0.63 | 80.99±3.38 | 75.83±1.30 | 78.35±0.64 | 83.26±1.13 | 74.18 | 6.9 |
| RGCL | 71.42±0.66 | 75.20±0.34 | 63.33±0.17 | 61.38±0.61 | 83.38±0.91 | 76.66±0.99 | 77.90±0.80 | 76.03±0.77 | 73.16 | 9.0 |
| D-SLA | 72.60±0.79 | 76.81±0.52 | 64.24±0.50 | 60.22±1.13 | 80.17±1.50 | 76.64±0.91 | 78.59±0.44 | 83.81±1.01 | 74.14 | 6.3 |
| SimGRACE | 71.25±0.86 | - | 63.36±0.52 | 60.59±0.96 | - | - | - | - | - | 11.3 |
| SEGA | 71.86±1.06 | 76.72±0.43 | 65.23±0.91 | 63.68±0.34 | 84.99±0.94 | 76.60±2.45 | 77.63±1.37 | 77.07±0.46 | 74.22 | 5.9 |
| GCS | 71.46±0.46 | 76.16±0.41 | 65.35±0.17 | 64.20±0.35 | 82.01±1.90 | **80.45±1.67** | 80.22±1.37 | 77.90±0.26 | 74.72 | 4.9 |
| LAMP-Mag | 74.72±1.24 | 76.86±0.31 | 64.92±0.55 | **64.85±0.73** | 85.18±2.12 | 78.91±2.55 | 80.38±0.75 | 84.72±1.77 | 76.32 | 2.3 |
| LAMP-Soft | **75.77±0.76** | **77.23±0.41** | **65.87±0.33** | 64.24±0.68 | **85.98±1.27** | 79.50±2.19 | **81.73±1.25** | **85.58±1.43** | **76.99** | 1.3 |

learning, GIN [45] with 32 hidden units and 3 layers is set up. In transfer learning, GIN is used with 5 layers and 300 hidden dimensions. The pruning ratio for sparse encoder is selected from 5% to 95% with a step of 5%. For local contrastive loss balance, $\alpha$ is tuned among $\{0.01, 0.1, 1, 10, 100\}$. Hyper-parameters are selected by the grid search on the validation sets. Additional details are shown in the Appendix C.

**Pruning strategy.** To demonstrate the compatibility of our framework, LAMP, with a variety of pruning methodologies, in this study, we adopt two distinct pruning techniques for the implementation of our method: magnitude pruning [12] and soft filter pruning [15], denoted as **LAMP-Mag** and **LAMP-Soft**, respectively.

## 5.2 Results Compared with SOTAs

**Unsupervised learning.** The baselines in unsupervised learning have three categories. The first set is three SOTA kernel-based methods that include GL [31], WL [30], and DGK [46]. The second set is four heuristic self-supervised methods, including node2vec [11], sub2vec [1], graph2vec [2], and InfoGraph [34]. The final category comes from the graph contrastive learning domain, including MV-GRL [13], GraphCL [51], AD-GCL [36], JOAO [50], AutoGCL [49], RGCL [24], SimGRACE [44], SEGA [41] and GCS [40].

The classification accuracies of LAMP against the SOTA competitors are shown in Table 1, and a significant performance improvement from the disappearance of the data augmentation can be witnessed as opposed to the baselines. Before the specific performance description of LAMP, here, we first glance at SimGRACE, which first attempts to transfer the contrastive attention from data augmentation to model perturbation via introducing Gaussian noise

to model weights. Although SimGRACE reveals its effectiveness on various datasets, the introduced Gaussian noise still degenerates the model performance; put differently, SimGRACE does not defeat the models with data augmentations.

We now present a comprehensive analysis of the superior performance of LAMP. As indicated by the final column for average rank, LAMP-Mag and LAMP-Soft secure the top two positions and exhibit the highest average accuracies among all baseline models. Notably, as evidenced by the column for average accuracy, LAMP-Soft surpasses the second-best model (i.e., GCS) with an accuracy improvement of 1.43%. Specifically, LAMP-Soft achieves the best performance on six out of eight benchmarks, while still maintaining the second-best performance on the remaining two datasets. Although LAMP-Mag is not superior as LAMP-Soft, it achieves the best performance on the NCI1 dataset and the second-best on three other datasets. Thus, we can conclude that model pruning may be a more promising direction for graph contrastive learning.

**Transfer learning.** Baseline methods in transfer learning include EdgePred, AttrMsking, ContexPred [17], Infomax [39], JOAO [50], GraphCL [51], AD-GCL [36], LP-Info [52], GraphMAE [16], Auto-GCL [49], GraphMVP [25], RGCL [24], D-SLA [19], SimGRACE [44], SEGA [41] and GCS [40]. A model without pre-train, termed 'No Pre-Train', is also adopted for comparison.

The results of LAMP, along with baselines under the setting of transfer learning on eight benchmarks, are shown in Table 2. In summary, the proposed models, namely LAMP-Mag and LAMP-Soft, demonstrate superior efficacy in comparison to preceding studies, as evidenced by the average ROC-AUC and ranking. Specifically, LAMP-Soft outperforms all other models on six out of the eight

**Table 3: The effectiveness of local contrastive loss $\mathcal{L}_{LocalC}$. A.A. is short for average accuracy.**

|          | SimGRACE    | LAMP-Mag w/o $\mathcal{L}_{LocalC}$ | LAMP-Mag    |
|----------|-------------|-------------------------------------|-------------|
| NCI1     | 79.12±0.44  | 80.26±0.58                          | 82.62±0.31  |
| PROTEINS | 75.35±0.09  | 75.73±0.75                          | 76.75±0.67  |
| DD       | 77.44±1.11  | 78.44±0.79                          | 79.47±0.97  |
| MUTAG    | 89.01±1.31  | 89.88±1.51                          | 90.02±1.59  |
| COLLAB   | 71.72±0.82  | 72.65±0.51                          | 74.62±0.75  |
| RED-B    | 89.51±0.89  | 90.28±0.49                          | 90.58±0.46  |
| RED-M5K  | 55.91±0.34  | 55.69±0.36                          | 56.42±0.26  |
| IMDB-B   | 71.30±0.77  | 71.68±0.67                          | 73.46±0.65  |
| A.A.     | 76.17       | 76.83                               | 77.99       |

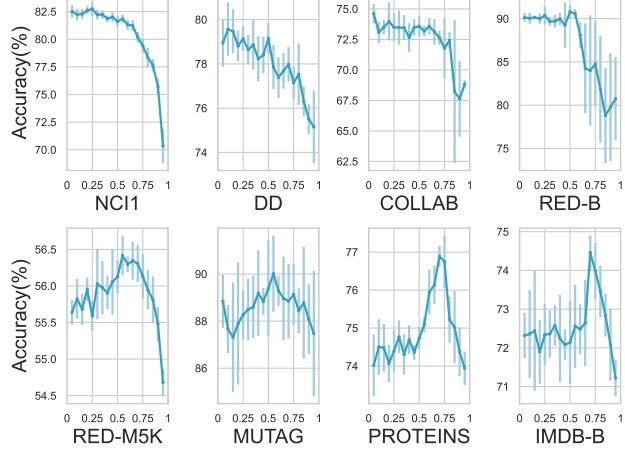

**Figure 4: Sensitivity *w.r.t.* pruning ratio $\gamma$.**

benchmarks, while securing second place on the remaining two datasets. LAMP-Mag exhibits optimal performance on the SIDER benchmark and obtains the second highest performance on five out of the eight benchmarks. In particular, LAMP-Soft obtains a 2.27% performance gain in terms of average ROC-AUC compared to the best baselines (i.e., GCS with a 74.72% average ROC-AUC) and over 10% performance gain compared to the model trained from scratch.

## 5.3 Ablation Study

Here, we make an in-depth analysis about the performance of LAMP under the setting of unsupervised learning. In particular, the magnitude pruning is adopted for ablation study.

**Effectiveness of local contrastive loss.** Besides the contrastive angle from the ubiquitous NT-Xent loss for unsupervised learning, we take another step to help model be capable of handling hard negative samples from the perspective of nodes. As shown in Table 3, LAMP-Mag obtains better performance with a 1.16% average accuracy gain when decorated with the proposed local contrastive loss, which suggests the effectiveness of the proposed local loss in addressing hard negative samples and improving model performance. In particular, LAMP-Mag w/o $\mathcal{L}_{LocalC}$ has an average accuracy of 76.83% on eight benchmarks, which not only outperforms current

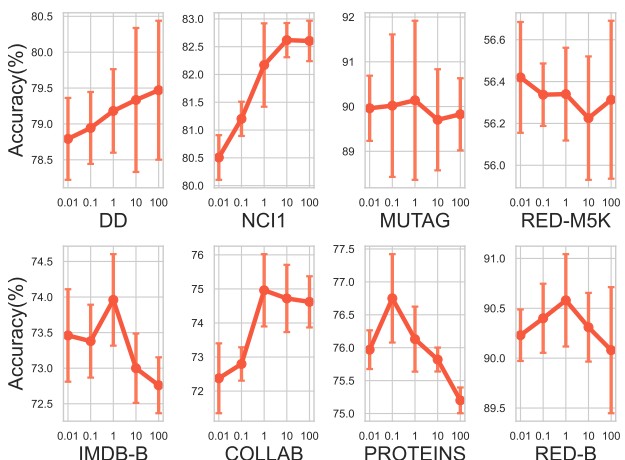

**Figure 5: Sensitivity *w.r.t.* loss balance $\alpha$.**

SOTAs for view generation but also defeats SimGRACE which disturbs the model weight with Gaussian noise. Thus, we can reaffirm the effectiveness of pruning on graph contrastive learning.

**Sensitivity regarding pruning ratio.** The pruning ratio controls the information that the sparse encoder captures; thus, a proper ratio would help the model identify the essential structure of input graphs. As shown in Figure 4, the datasets in the first row prefer a lower pruning ratio, while the other datasets would like a sparser encoder. Moreover, regardless of the optimal pruning ratio, the performances of all datasets quickly deteriorate when the sparsity goes above 75%, due to limited capacity.

**Sensitivity regarding loss balance.** As we have validated the effectiveness of the proposed local contrastive loss, we further inspect the influence of its hyper-parameter (i.e., $\alpha$) on model performance. The unsupervised results of LAMP-Mag with candidate $\alpha$ on eight benchmarks are shown in Figure 5. As can be seen, DD and NCI1 show a positive correlation with $\alpha$, while MUTAG is not sensitive to the choice of $\alpha$, which is consistent with the stable performance of MUTAG w/o $\mathcal{L}_{LocalC}$ in Table 3. The other datasets show a trade-off within the given selections, and generally obtain the best performance with $\alpha$ around 1.

## 6 CONCLUSION

In this work, we reformulate the problem of graph contrastive learning from the angle of model compression. To avoid the loss of semantics caused by data augmentation, we present a novel method based on model pruning, termed *LAMP*, rather than relying on profound domain knowledge. Before the empirical validation, we theoretically explain the superiority of model pruning compared to data augmentation. Extensive experiments under unsupervised and transfer learning show that LAMP suppresses the current SOTA methods based on data augmentations. An automatic pruning ratio and more advanced pruning strategies shed light on the future research direction.

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
