# OpenReview forum: "Uncovering Capabilities of Model Pruning in Graph Contrastive  Learning"
_acmmm.org/ACMMM/2024/Conference — MM2024 Poster_

### Official Review · Reviewer_pcvK · 2024-05-25

**Rating:** 3
**Confidence:** 3

**Summary:**

This paper introduces LAMP, a novel graph contrastive learning framework that uses model pruning instead of data augmentation to enhance generalization and avoid semantic alteration. This approach theoretically and practically outperforms existing methods on various graph classification benchmarks, both in unsupervised and transfer learning scenarios. Additionally, LAMP incorporates a local contrastive loss to effectively handle hard negative samples, demonstrating significant improvements over state-of-the-art models in terms of accuracy and robustness.

**Strengths:**

Strengths:

1.	The idea of using model pruning rather than traditional data augmentation for graph contrastive learning is novel. This concept not only preserves the semantic integrity of the data but also broadens the applicability of graph neural networks without relying on domain-specific augmentations.

2.	The paper provides a strong theoretical basis for the proposed method. It includes a theoretical analysis demonstrating the advantages of model pruning over data augmentation in maintaining the structural and semantic properties of graphs, which adds depth and credibility to the research.

3.	The experiments are extensive and well-structured, covering a variety of benchmarks and scenarios. The results clearly demonstrate the superiority of the LAMP framework over state-of-the-art methods, offering empirical evidence that supports the theoretical claims.

**Limitations:**

1.	This paper lacks a discussion on the scalability issues, the computational overhead of pruning strategies, and the impact of pruning on very large graphs, which might help in setting realistic expectations for its applicability.

2.	The method's success is heavily dependent on the choice of pruning techniques. Although this allows flexibility, it might also lead to inconsistencies in performance across different applications or datasets, depending on how the pruning is implemented.

3.  While the paper claims superiority over data augmentation methods, it could provide a more direct comparative analysis, demonstrating how model pruning performs in scenarios where data augmentation has traditionally excelled.

**Suitability:**

2

---

### Official Review · Reviewer_WcYq · 2024-05-29

**Rating:** 3
**Confidence:** 4

**Summary:**

This paper introduces LAMP (Learning via Model Pruning), a novel framework that applies model pruning to graph contrastive learning, deviating from traditional data augmentation methods to preserve semantic integrity. By dynamically generating a perturbed graph encoder and contrasting it with the original through the pruning of transformation weights, LAMP avoids domain-specific dependencies and improves handling of hard negative samples with a local contrastive loss. Extensive testing demonstrates LAMP's superiority over state-of-the-art models in both unsupervised and transfer learning for graph classification.

**Strengths:**

1. The paper benefits from clear, concise, and well-structured writing. The authors effectively communicate complex ideas and methodologies in a manner that is accessible to readers, which enhances the paper's overall impact and readability. This clarity extends to the presentation of experimental results and theoretical concepts, making the novel approach understandable and the results compelling.
2. The framework is not only theoretically grounded with detailed analyses of its superiority over data augmentation but is also rigorously tested in practical scenarios. This dual approach strengthens the credibility and reliability of the proposed method.

**Limitations:**

1. While the paper successfully demonstrates the advantages of LAMP over many baselines, it does not extensively explore or compare against other novel augmentation alternatives in the same experimental setting that could potentially offer similar benefits. This limits the scope of the study and may overlook other viable methods in graph contrastive learning.
2. The paper focuses primarily on graph classification tasks, which, while providing a robust test bed for the proposed methodology, does not discuss the potential implications or adaptability of the LAMP framework to other types of graph-related tasks (like node classification and link prediction) or broader application domains. This could be seen as a missed opportunity to highlight the versatility of the approach.

**Suitability:**

2

---

### Official Review · Reviewer_C61H · 2024-06-06

**Rating:** 5
**Confidence:** 3

**Summary:**

The paper presents a framework for enhancing graph contrastive learning by focusing on model pruning rather than traditional data augmentation techniques. The authors argue that typical data augmentations can alter the semantic integrity of the data and propose using model pruning as an alternative to preserve semantic information and improve model generalization.

The proposed methods operate by dynamically pruning a graph neural network to create contrasting versions of the model, which are then used to generate embeddings for contrastive learning. This approach not only preserves the original graph's semantic information but also enhances the learning process by addressing the issue of hard negative samples through a local contrastive loss.

The paper provides theoretical insights and conducts extensive experiments to demonstrate LAMP's strength over the other baselines on the graph classification tasks.

**Strengths:**

The shift from traditional data augmentation to model pruning is novel in the context of graph contrastive learning.

Extensive experiments are conducted to show the proposed methods' superior performance over the baselines across numerous benchmarks on the graph classification tasks for both unsupervised/transfer learning, making the generalizability of LAMP convincing.

The paper provides theoretical insights for its proposed method.

The introduction of the local contrastive loss has a reasonable performance boost.

**Limitations:**

The proof of theorem 4.1 does not seem to be rigorous enough. For example, Equation 9 relies on the conclusion given by https://arxiv.org/pdf/2002.00585, but in that paper, they made many assumptions such as over-parameterization, has the paper checked those assumptions carefully?

The performance gain looks over-rely on the local contrastive loss. A direct comparison between Table 3 and Table 1 shows that if we remove the local contrastive loss, the performance gain will be very marginal.

Combining Figures 4 and 5, it looks like a good pruning ratio is relatively important for datasets like PROTEIN, RED-M5K, and IMDB-B, whereas datasets like NCI1 and DD rely on local contrastive loss more. However, the paper does not give any qualitative or qualitative analysis about the nature of those datasets to explain why that is the case.

It would be great to include some hard-negative examples like those in Figure 3 from the experiment datasets as an ablation study.

The related work section is missing some citations for other graph contrastive learning works.

**Suitability:**

2

---

### Official Review · Reviewer_1YYd · 2024-06-09

**Rating:** 5
**Confidence:** 2

**Summary:**

This work presents a view of graph pruning based pre-training by graph contrastive learning via contrasting different model versions rather than augmented views. The authors provide interesting theoretical proof to show the effectiveness of model pruning. Then, they use the original graph as input and dynamically generate a perturbed graph encoder to contrast with the original encoder. In addition, a local contrastive loss is proposed to handle the hard negative samples during the training. The proposed method achieves SOTA performance on various graph benchmarks.

**Strengths:**

1, The idea of using pruning-based method for graph contrastive learning  is interesting and novel to me.

2, The overall writing is good and easy to follow.

3, I check the proof part and find it interesting and convincing, which motivates the research using model pruning.

4, The proposed methods achieve stronger performance .

**Limitations:**

1, The comparison with previous augmentation-based graph contrastive pre-training is week. I think more detailed comparisons are needed.

2, The scope problems. I wonder whether the submission meets the requirements of MM since there is only graph data and no multi-modal data involved.

3, Several presentation issues:
	Missing error analysis.
	Missing training and inference cost analysis.

**Suitability:**

2

---

### Meta-Review · Area_Chair_51di · 2024-07-01

**Recommendation:** Accept (Poster)
**Confidence:** 4

**Metareview:**

The authors have adequately addressed the concerns raised by most reviewers and received three positive ratings and one negative rating. As reviewer WcYq did not provide a final rating after the rebuttal, most reviewers agree that this paper is ready for publication. Therefore, the paper is recommended for acceptance.